# A System Based-Approach to Examine Host Response during Infection with Influenza A Virus Subtype H7N9 in Human and Avian Cells

**DOI:** 10.3390/cells9020448

**Published:** 2020-02-15

**Authors:** Biruhalem Taye, Hui Chen, Dawn Su-Yin Yeo, Shirley Gek-Kheng Seah, Michelle Su-Yen Wong, Richard J Sugrue, Boon-Huan Tan

**Affiliations:** 1School of Biological Sciences, Nanyang Technological University, 60 Nanyang Drive, Singapore 637551, Singapore; bbeyene@embl-hamburg.de (B.T.); chenh1@gis.a-star.edu.sg (H.C.); 2Bioinformatics Institute, 30 Biopolis Street, #07-01 Matrix Building, Singapore 138671, Singapore; 3European Molecular Biology Laboratory, Notkestraβe 85, 22607 Hamburg, Germany; 4Genomic Institute of Singapore, 60 Biopolis Street, Genome #02-01, Singapore 138672, Singapore; 5DSO National Laboratories, Defence Medical and Environmental Research Institute, 27 Medical Drive, Singapore 117510, Singapore; yeodawn@dso.org.sg (D.S.-Y.Y.); sgekkhen@dso.org.sg (S.G.-K.S.); wsuyen@dso.org.sg (M.S.-Y.W.); 6Infection and Immunity, Lee Kong Chian School of Medicine, Nanyang Technological University, 11 Mandalay Road, Singapore 308232, Singapore

**Keywords:** Influenza A virus, H7N9, A549 cells, CEF cells, differentially expression genes (DEGs), HIPPO signaling pathway

## Abstract

Although the influenza A virus H7N9 subtype circulates within several avian species, it can also infect humans with a severe disease outcome. To better understand the biology of the H7N9 virus we examined the host response to infection in avian and human cells. In this study we used the A/Anhui/1/2013 strain, which was isolated during the first wave of the H7N9 epidemic. The H7N9 virus-infected both human (Airway Epithelial cells) and avian (Chick Embryo Fibroblast) cells, and each infected host transcriptome was examined with bioinformatic tools and compared with other representative avian and human influenza A virus subtypes. The H7N9 virus induced higher expression changes (differentially regulated genes) in both cell lines, with more prominent changes observed in avian cells. Ortholog mapping of differentially expression genes identified significant enriched common and cell-type pathways during H7N9 infections. This data confirmed our previous findings that different influenza A virus subtypes have virus-specific replication characteristics and anti-virus signaling in human and avian cells. In addition, we reported for the first time, the new HIPPO signaling pathway in avian cells, which we hypothesized to play a vital role to maintain the antiviral state of H7N9 virus-infected avian cells. This could explain the absence of disease symptoms in avian species that tested positive for the presence of H7N9 virus.

## 1. Introduction

Influenza viruses are an important cause of respiratory infection, and the global disease burden for influenza infections is estimated by the World Health Organisation to be responsible for up to 650,000 deaths annually (http://www.who.int/mediacentre/news/releases/2017/seasonal-flu/en/). Influenza viruses are members of the *Orthomyxoviridae,* and based on antigenic differences in the nucleoprotein (NP) and matrix (M) proteins the influenza viruses can be classified into four types called A, B, C and D. Influenza A viruses can be further subtyped based on the antigenicity of the haemagglutinin (HA) and neuraminidase (NA) surface glycoproteins, giving rise to 18 HA (H1 to H18) and 11 NA (N1 to N11) subtypes. They have a wider host range than the other influenza virus types, and have been isolated from humans and a variety of different animal species (e.g., birds, pigs, marine mammals). Influenza A virus strains are maintained in aquatic bird populations which are believed to be an important natural reservoir for the influenza A virus strains that infect all other animal species and humans [1,2,3]. In the context of human infections, many regions of the world experience seasonal epidemics involving increased human-to-human transmission of influenza virus and disease burden. These human-adapted viruses are often referred to as seasonal influenza virus, and in the Northern and Southern hemispheres, the circulating influenza virus strains that predominate can vary. Previous influenza pandemics have involved influenza viruses that were transmitted from birds, into swine, and then to humans. Evidence suggests that swine acts as an intermediate species [4], enabling the adaptation of avian-origin viruses to an alternative mammalian host prior to infecting humans. The capacity of influenza viruses to evolve and adapt to replicate in these different animal hosts is directly related to their capacity for interspecies transmission. Although it was originally thought that the transmission of avian influenza virus to humans could only occur via an interspecies host (e.g., swine), avian influenza viruses (e.g., H5N1) can also be transmitted directly from birds to humans. However, such events are usually self-limiting, since these viruses do not adapt to efficiently replicate in a mammalian host, and they do not exhibit efficient human-to-human transmission. Influenza virus evolution is the driver for influenza virus interspecies adaptation and transmission, and this is mediated by the high mutation rates and reassortment of genomic segments between two or more influenza viruses. In a relatively small time-scale, new virus variants can be potentially generated, and these two processes have been the basis for past influenza virus pandemics [5,6,7].

The capacity of an avian influenza virus to adapt to mammalian host is dependent upon several factors (reviewed in [8]). Specific amino acid sequence motifs that are associated with host adaptation have been identified within several different virus proteins. In some specific cases, biological functions associated with sequence-specific motifs have been proposed [9,10]. In general, the role that these sequence motifs play in mediating species adaptation is poorly defined. Since a significant degree of sequence variation exists among different avian influenza viruses, this is likely to influence the molecular process that leads to host adaptation. As a consequence, host adaptation is expected to be both multifactorial, and to some extent, virus strain-specific.

The avian influenza viruses of the subtypes H5 and H7 have the capacity to convert into highly pathogenic avian influenza (HPAI) viruses, which are associated with high mortality rates. Although some specific correlates that lead to the emergence of HPAI viruses have been identified, the underlying mechanism for the predisposition for a low pathogenic avian influenza (LPAI) virus to convert to HPAI virus is unclear. LPAI H9N2 virus strains are widespread, and they are mainly associated with poultry disease, and H9N2 virus infection in humans leads to relatively mild symptoms [11]. The H9N2 virus can infect pigs, and these are believed to be the intermediate species in many avian-to-human transmission events. Interestingly an H7N9 virus strain was described in 2013 that was responsible for significant disease severity in humans [12,13], and this virus contained six internal genes that originated from circulating H9N2 viruses [14,15,16]. The H7N9 virus first emerged in China’s Yangtze River Delta in March 2013 [17], and since the initial detection, there have been five waves of infection that have been associated with increased mortality rates [18,19]. In human infections, the H7N9 virus showed a bias towards male infection with age older than 50 years old, suggesting that human host factors play a role in biological susceptibility to the virus infections. The last wave of H7N9 virus infection was also associated with the emergence of an HPAI virus variant which was implicated as the causative virus for several human-to-human clusters. The H7N9 seemed to show unusually greater transmissibility, and higher mortality rates with a more severe disease outcome in humans than any other H7 viruses reported to date.

Our previous studies on different influenza A virus subtypes with human lung airway epithelial (A549) cells and chick embryo fibroblast (CEF) cells have suggested virus-specific replication characteristics and anti-virus signaling [20]. In this study, we infected both A549 and CEF cells with the H7N9 virus, and performed a system-based approach to compare the infected host transcriptome with other previously reported influenza A virus subtypes. Bioinformatics tools were applied to annotate probes and human ortholog mapping (*Gallus gallus* versus *Homo sapiens*) as described [21]. Ortholog mapping of host genes identified significant down-regulation of several pathways between H7N9 and another influenza A virus subtypes, in infected A549 and CEF cells. This agreed with our previous findings that the replication characteristic of different influenza A virus subtypes in infected A549 or CEF cells, are unique and virus-specific.

## 2. Materials and Methods

### 2.1. Viruses, Cells, Antibodies

Influenza A virus subtype H1N1 strain WSN [A/WSN/1933 (V-1520)] was purchased from American Type Culture Collection (ATCC), USA. The subtype H7N9 virus (A/Anhui/1/2013) [17] was a gift from Dr. Ian Barr, WHO Collaborating Centre for Influenza, Melbourne. (A/Duck/Malaysia/F59/2004; A/Duck/Malaysia/F118/2004; A/Duck/Malaysia/F189/2004) and H5N3 (A/Duck/Singapore-Q/F119/1997) and H9N2 (A/Duck/Malaysia/02/2001) viruses were obtained from Agri-Food and Veterinary Authority of Singapore and were characterized previously [22]. The viruses were propagated in 9 to 11 day old embryonated chicken eggs using standard protocols. UV inactivated virus was prepared at 4°C by exposing the virus inoculum (at a distance of 1cm) to a UV radiation source (λ = 256 nm) for 30 mins. Madin-Darby canine kidney (MDCK, ECACC 84,121,903) cells used for median tissue culture infectious dose (TCID) titration and the human alveolar basal epithelial (A549, ECACC 86,012,804) for the experiments, respectively, were purchased from European Collection of Authenticated Cell Cultures (ECACC), and maintained in Dulbecco’s Modified Eagle’s Medium (DMEM) (Invitrogen, Carlsbad, CA, USA) containing 10% fetal bovine serum (FBS) (Invitrogen, USA) and 1% penicillin/streptomycin (pen/strep) (Invitrogen, USA). Chick embryo fibroblasts (CEF) were prepared from 8 to 10 day-old chick embryos and maintained in DMEM with 10% FBS and 1% (pen/strep) as described in [20]. Virus infections in both A549 cells and CEF cells were carried out in DMEM in 2% FBS, at 37 °C in the presence of 5% CO_2_. For immunofluorescence microscopy, the anti-NP (Chemicon, San Diago, CA, USA) was purchased. For western blotting, the primary antibodies to STAT1 and pSTAT1 were purchased from BD Transduction Technology, USA; and the anti-mouse or anti-rabbit IgG (whole molecule) peroxidase conjugate from Sigma-Aldrich, St Louis, MI, USA. All experiments involving H7N9 viruses were performed in biosafety level 3 facility according to the Biological Agents and Toxin Act, Ministry of Health, Singapore.

### 2.2. Immunofluorescence Microscopy

A549, MDCK and CEF cells were seeded onto 13 mm glass cover and infected with each influenza virus subtype at multiplicity of infection (MOI) of 5 as described in [20]. At 2, 4, 6, 8, and 10 hours post-infection (hpi), the cells were fixed with 4% (*w/v*) paraformaldehyde (Sigma-Aldrich, USA) in phosphate-buffered saline (PBS) and permeabilized in 0.1% (*w/v*) saponin (Sigma-Aldrich, USA) in PBS. The cells were labeled with anti-NP and the stained cells were mounted onto slides using Dakocytomation (Dako, Santa Clara, CA, USA) and visualized using immunofluorescence microscope (Model BX51, Olympus, Parkway Center Valley, PA, USA).

### 2.3. Quantitative PCR to Measure Copies of Host Genes

A549 cells were infected with each influenza virus subtype MOI of 5. At 10 hpi, total RNA was extracted from cells at 4 °C using the Rneasy kit (Qiagen, Hilden, Germany) and reverse-transcribed using Superscript II (Invitrogen, USA) according to manufacturer’s instructions. Quantitative Real-time PCR (qPCR) for the selected host genes was carried out in duplicate reactions as described in [20], with the iCycler System (BioRad, Hercules, CA, USA).

### 2.4. Western Blot Analysis

Cell lysates were prepared in 1X boiling mix (1% SDS, 15% glycerol, 1% β-mercaptoethanol, 60 mM sodium phosphate, pH 6.8) and heated at 100 °C for 2 min. The protein sample was separated by SDS-PAGE. After SDS-PAGE, the proteins were transferred on to PVDF membranes using the mini blotting apparatus (BioRad, USA), after which the membranes were washed with PBSA and blocked for 18 h at 4 °C in PBSA containing 1% BSA and 0.05% Tween 20. The membrane was incubated with the respective specific primary antibody to STAT1 and pSTAT1, followed by the appropriate anti-mouse or anti-rabbit IgG (whole molecule) peroxidase conjugate (Sigma, USA). The protein bands were visualized using the ECL protein detection system (Amersham, Buckinghamshire, UK). In all cases, the apparent molecular masses were estimated using Kaleidoscope protein standards (BioRad, USA).

### 2.5. Microarray Experiment

A549, and CEF cells were either mock-infected or infected with each virus subtype at MOI of 5. At 2 and 10 hpi, the cells were harvested at 4 °C using RNAlater (Ambion, Carlsbad, CA, USA) diluted in 1:1 with PBS buffer, aliquoted, pelleted and stored at −80 °C. Total RNA was extracted from approximately 1 × 10^7^ cells using Rneasy mini kit (Qiagen) and quantified using the Nanodrop ND-1000 Spectrophotometer (ThermoFischer Scientific, Massachusetts, USA). Double-stranded cDNA was synthesized from 3 µg of total RNA with the GeneChip One-cycle cDNA synthesis kit (Affymetrix, Santa Clara, CA, USA), followed by synthesis of biotin-labeled cRNA using the GeneChip IVT labeling kit (Affymetrix, USA), according to manufacturer’s instructions. After cRNA fragmentation, 15 µg of biotin-labeled cRNA from virus-infected A549 and CEF cells was hybridized to the Human U133 2.0 Genome Array and GeneChip Chicken genome Array (Affymetric, USA), respectively. Three independent experiments were performed, and the data analyzed separately. The arrays were washed and stained using the Hybridization, Wash and Stain Kit (Affymetric, USA) and the GeneChip Fluidics Station 450 (Affymetrix, USA) according to standard Affymetrix protocols. Finally, the arrays were scanned with the GeneChip scanner 3000 (Affymetrix, USA). Quality control, GeneChip hybridization and data acquisition were performed according to the standard protocols available from Affymetrix.

### 2.6. Data Analysis and Functional Annotations

Affymetrix CEL files were generated from GeneChip Operating Software (GCOS) version 5.0 (GCOS, Affymetrix, USA) and subsequently imported to AltAnalyze 2.1.0 (Cincinnati, OH, USA) for analysis [23]. Robust multichip average (RMA) method was used to normalize all Affymetrix CEL files. The normalized expression data were clustered using Principal component analysis (PCA). The limma (R package) based moderated t-test, ANOVA, Benjamini-Hochberg False Discovery and log^Fold changes (FC)^ were calculated to identify genes with statistically significant changes between the virus-infected and control (mock) samples. Probes with the Benjamini-Hochberg False Discovery (adjusted *p*-value) < 0.05 and |log^FC^| > 1 were considered differentially expressed. A single gene might be represented by more than one probes, hence, those genes with an adjusted *p*-value < 0.05 and their probe mean |log ^FC^| > 1 were considered as differentially expressed genes (DEGs). The DEGs in at least one virus at one-time point infection with their corresponding log^FC^ values were used for average linkage hierarchical clustering using Gene Cluster 3.0 [24]. The heat maps of the clustering were viewed by Java Tree Viewer (version 3.0, Princeton, NJ, USA). Venn diagrams showing the overlap of DEGs in different groups were created using Venny 2.1 [25]. To compare the expression changes of the H7N9 infections in A549 and CEF cells, we have used our previous probe annotation and human ortholog mapping (*Gallus gallus* vs. *Homo sapiens*) [21], and the human orthologs genes were used for further analysis. In addition, the expression data from our previous work (A/WSN/1933 (H1N1/WSN) (VR-1520), A/Duck/Malaysia/F118/2004 (H5N2/F118), A/Duck/Malaysia/F189/2004 (H5N2/F189), A/Duck/Malaysia/F59/2004 (H5N2), A/Duck/Singapore-Q/F119/1997 (H5N3) and A/Duck/Malaysia/02/2001 (H9N2) virus infections in A549 and CEF cells) [20,21] was processed in a similar manner in order to be compared with the expression changes during H7N9 virus infections. The functional enrichment analysis of the DEGs were analyzed using Gene Annotation (ncbi.nlm.nih.gov) and Analysis Resource-Metascape [26], and minimum overlap of three genes and *q*-value < 0.01 was used to select the top 20 statistically significant pathways. All microarray data were deposited in NCBI Gene Expression Omnibus (GEO) DataSets (ncbi.nlm.nih.gov) with accession number GSE31524.

## 3. Results

### 3.1. Examination of A549, CEF and MDCK Cells Infected with Influenza A Virus Subtype H7N9

We used the A549 and CEF cell lines to examine H7N9 virus infection, since these are commonly cell model systems to examine influenza virus infection in human and avian cell backgrounds respectively. A549 and CEF cells were infected with influenza A virus subtype H7N9 using an MOI of 5 and at 2, 4, 6, 8, and 10 hpi the virus-infected cells were stained with anti-NP and examined by immunofluorescence (IF) microscopy (Figure 1A,B). The antibody targeting the NP was selected for this study as it reacted with all influenza A virus subtypes, as reported in our previous host response studies [20]. At 4 hpi, we observed a low level of anti-NP staining in the virus-infected A549 cells, which appeared to be localized in the nucleus. The anti-NP staining intensity increased up to 10 hpi, with greater than 95 % of the cells showing anti-NP staining. Anti-NP staining in virus-infected CEF cells was only detected from 6 hpi onwards, but at 10 hpi greater than 95% of the cells showed anti-NP staining. At 10 hpi anti-NP staining was observed in both the nucleus area and the cytoplasm, and the staining was consistent with efficient nuclear export of the RNP complexes in both A549 and CEF cells infected with the H7N9 virus. The H7N9 virus was either untreated or irradiated with UV light, and then used to infect A549 and MDCK cells. MDCK cells are susceptible to all influenza virus subtypes and are commonly used to study influenza virus infection. The anti-NP stained cells were examined using IF microscopy and anti-NP staining was only observed in the cells infected with the untreated virus (Figure 1C). The absence of anti-NP staining confirmed the inactivation of the H7N9 virus by UV irradiation, and indicated that this would be an appropriate control with which to examine the host response to infection using the microarray analysis. Our analysis also indicated that a time-point of infection at 2 and 10 hpi would be suitable times to compare the host cell response to H7N9 virus infection at early and late times of infection in these cells. In addition, it suggested that 10 hpi would be suitable time to compare host responses in these cell lines with other representative avian influenza subtypes.

### 3.2. Differentially Expressed Genes (DEGs) in H7N9 Virus-Infected A549 Cells and Their Functional Annotations

The host response for H7N9 virus-infected-A549 cells was examined at the global gene expression level using the GeneChip human genome HG U133 2.0 Genome array (Affymetrix). A549 cells were infected with untreated and UV-treated H7N9 virus, and at 2 and 10 hpi, the infected host transcriptome was analyzed. The gene expression data in H7N9 virus-infected A549 cells was normalized, and principal component analysis (PCA) was used to cluster the expression changes between samples (Figure 2A). The gene expression changes in A549 cells infected with the virus at 2 hpi, UV-treated virus at 2 hpi and mock-infections, were observed to cluster together. In contrast, at 10 hpi, the gene expression changes in A549 cells infected with the virus and UV-treated virus showed different clustering, suggesting that the gene expression changes are distinctly unique. The gene expression changes were compared with mock-infected A549 cells, and a total of 3517 and 718 genes were differentially expressed in untreated and UV-treated H7N9 virus infections, respectively, at 10 hpi (Figure 2B). The DEGs in untreated virus infections were four times or higher than the UV-treated infections. In addition, pair-wise analysis of the DEGs at the 10 hpi showed 207 up-and 249-down-regulated genes shared commonly between untreated and UV-treated virus infections (Figure 2C). No DEGs were detected at 2 hpi with untreated H7N9 virus infections. These findings further confirmed that the gene expression changes were replication dependent.

We selected specific host genes in cytokine expression and interferon (IFN) signaling (IFNβ1, IL28a, CCL5), and IFN-stimulated expression [2′,5′-oligoadenylate synthase (OAS2), Myxovirus 1 (MX1), RSAD2 (radical S-adenosyl methionine domain containing 2) that are commonly up-regulated in influenza virus-infected cells for further study. Their FC in gene expression using microarray analysis was confirmed by using qPCR targeting the respective host gene (normalization to the housekeeping gene, EF) (Appendix A, Appendix A). The CCL5 gene expression changes were observed to have the highest FC for both microarray and qPCR data; followed by IL28a and RSAD2. Although immunoblot analysis revealed reduced levels of the Signal Transducer Activator of Transcription 1 (STAT1) protein in H7N9 virus-infected cells, the presence of phosphorylated STAT1 (pSTAT1) in H7N9 virus-infected A549 cells was consistent with the induction of IFN expression. The pSTAT1 protein was not detected in mock-infection or infection with UV-treated virus. Taken together, our findings indicated that infection of A549 cells with the H7N9 virus can cause significant global gene expression changes, with the most significant changes observed at 10 hpi.

Since the 10 hpi time point displayed the most gene expression changes, the DEGs at this time point was used for functional annotation (KEGG pathway enrichment analysis). Generally, functional annotations of the DEGs relating to cell cycle, cellular organization, metabolic processes and immune response pathways were observed, and the top 20 most significant pathways are shown in Figure 2D,E and Appendix A. Pathways relating to influenza A virus and P53 signaling were the most enriched by up-regulated genes for virus infections when compared with UV-treated (Figure 2D). Other pathways, such as Janus Kinase (JAK)-STAT signaling pathway, Tumor Necrosis Factor (TNF) signaling pathway and cytokine-cytokine receptor interactions were exclusively up-regulated pathways in virus infection. Induction of these pathways suggested that they were stimulated by an active viral replication in virus-infected A549 cells. In contrast, many metabolic pathways and cell cycle processes were down-regulated during virus infections, while both virus and UV-treated infections commonly down-regulated peroxisome pathway (Figure 2E).

### 3.3. DEGs in H7N9 Virus-Infected CEF Cells and Their Functional Annotations

The host response for H7N9 virus in CEF cells was next examined in the same manner as with the virus-infected A549 cells, at the global gene expression level using the GeneChip Chicken genome Array (Affymetrix). CEF cells were infected with untreated and UV-treated H7N9 virus, and at 2 and 10 hpi, the infected host transcriptome was analyzed. The PCA of normalized data indicated distinct clusters of expression changes in UV-treated, untreated and mock-infected cells (Figure 3A). The gene expression changes in CEF cells infected with untreated and UV-treated H7N9 virus at 2 hpi, were observed to cluster together. Similar to the findings in virus-infected A549 cells, unique and higher expression changes were significantly observed in virus-infected CEF cells at 10 hpi when compared with UV-treated virus infections. The gene expression changes in CEF cells infected with the virus and UV-treated virus at 10 hpi showed different clustering, suggesting that the gene expression changes were distinctly unique. The gene expression changes were compared with mock-infected CEF cells, and a low number of genes were differentially expressed in untreated (24 genes) and UV-treated H7N9 virus (19 genes) infections, respectively, at 2 hpi (Figure 3B). At 10 hpi, a higher number of genes were differentially expressed in untreated (4626 genes) and UV-treated H7N9 virus (874 genes) infections, respectively. At this same time point of infection, there were 209 up-regulated and 309 down-regulated genes commonly shared between untreated and UV-treated virus infections (Figure 3C). Taken together, the findings showed that significant gene expression changes are only observed at 10 hpi and were replication-dependent.

The KEGG pathway analysis showed that the up-regulated genes in virus-infected CEF cells were unlike the finding in virus-infected A549 cells in that there were several pathways commonly observed between the untreated and UV-treated virus infections (Figure 3D,E and Appendix A). However, HIPPO, MAPK and TGF-β signaling pathways were more enriched in the untreated virus-infected CEF cells, suggesting that these pathways were induced during active H7N9 infection. Pathways relating to cell cycle were down-regulated by both untreated and UV-treated virus infections. There were unique metabolic pathways down-regulated in the untreated H7N9 virus-infected CEF cells, which were also observed as being down-regulated in the virus-infected A549 cells. In addition, pathways relating to the lysosome and protein processing in the endoplasmic reticulum were highly enriched with down-regulated genes in the untreated virus-infected CEF cells.

### 3.4. Comparison of DEGs in H7N9 virus-infected A549 and CEF cells

Taken together, our findings suggested that infection of A549 and CEF with H7N9 virus could cause significant global gene expression changes, with the most significant changes observed with down-regulated gene expression changes at 10 hpi. To compare the gene expression changes in virus-infected CEF cells to that in A549 cells, orthology mapping was performed and the DEGs compared and shown in Figure 4. As expected, there were higher expression gene changes during infections with the untreated H7N9 virus when compared with UV-treated virus. The gene expression changes in virus-infected A549 cells clustered away from that of virus-infected CEF cells (Figure 4A). Mock-infections for both A549 and CEF cells clustered together. Pairwise analysis of the DEGs showed that at 10 hpi, there were more replication-dependent up-regulated genes in virus-infected CEF cells (1548 genes) than with virus-infected A549 cells (716 genes) (Figure 4B,C). There were 184 common replication-dependent up-regulated genes shared in both virus-infected A549 and CEF cells (Figure 4C). Similarly, there were 555 common ortholog genes that were down-regulated during active replication.

The KEGG pathway analysis of the up-regulated ortholog genes showed that several pathways were shared between the virus-infected A549 and CEF cells, but their induction was displayed with different magnitude (Figure 4D). For example, at 10 hpi, influenza A and RNA degradation pathways were more prominent in virus-infected A549 cells, whereas, TGF-β signaling pathways were more prominent in virus-infected CEF cells. There were also pathways induced that were unique to each virus-infected cell type. HIPPO signaling pathway was very prominent and unique to virus-infected CEF cells. Unlike A549 cells, the UV-treated virus was also able to induce the activation of many pathways in CEF cells. Ferroptosis pathway was commonly observed in both UV-treated virus-infected cell types, while p53 signaling pathway was common to both virus and cell infections. The functional annotation of the down-regulated ortholog genes indicated that down-regulation of several metabolic pathways were an indication of active H7N9 virus replication in both cell types (Figure 4E).

To compare the gene expression changes of the H7N9 virus-infected A549 and CEF cells at 10 hpi further, the ortholog replication-dependent genes were integrated with 1561 proviral and 152 antiviral host factor genes manually curated from published siRNA screening studies [21]. Overall, 67 and 107 proviral genes were up-regulated, respectively, in virus-infected A549 and CEF cells (Figure 5A). Among these up-regulated proviral genes, 21 genes were found in common between the two cell types and their expression levels were represented by the heat map (Figure 5B). In contrast, there were lesser number of antiviral genes that were up-regulated in both virus-infected cell types. There were 11 and 13 antiviral genes observed in H7N9 virus-infected A549 and CEF cells, respectively (Figure 5C), and only two genes (Figure 5B, highlighted in red) commonly observed in both cell types. The functional annotation of the replication-dependent up-regulated ortholog genes with respect to the proviral and antiviral host factors confirmed that TGF-β signaling pathway was found to be the common proviral pathway activated in both A549 and CEF cells during H7N9 virus infection (Figure 5C). In addition, the HIPPO signaling pathway was found to be a unique proviral pathway activated in the virus-infected CEF cells.

### 3.5. Comparison of DEGs with Other Influenza A Virus Subtypes

The expression changes of ortholog genes of H7N9 virus-infected A549 cells and CEF cells were compared with the infections of other influenza A virus subtypes (H1N1/WSN, H5N2, H5N3 and H9N2) in the same cell types. The comparison resulted in a hierarchical clustering of the DEGs, and these are represented in the heat map, shown in Figure 6A. The results indicated that when compared with other virus subtypes, the expression changes in H7N9 virus-infected cell type are unique, and higher expression changes were induced in both the H7N9 virus-infected CEF and A549 cells. When the KEGG pathways were analysed for each virus subtype at 10 hpi, several pathways were activated during H7N9 virus infections (Figure 6B,C). However, only the proviral HIPPO signaling pathway was activated in H7N9 virus-infected CEF cells. In addition, the H7N9 virus infection down-regulated several more metabolic pathways in A549 and CEF cells that were observed to be uniquely different from the other virus subtypes. Surprisingly, there were no significant down-regulated genes observed in the H9N2 virus-infected CEF at 10 hpi.

## 4. Discussion

Influenza A viruses can bind to 2 types of host receptors which are linked to sialic acid (NeuAc): human-type NeuAc α2,6 galactose are found predominantly on mammalian cells and avian-type NeuAc α2,6 galactose on avian cells; whereas, the canine cell line contains both types of receptors. H7N9 virus can bind to both types of receptors found in Human Airway Epithelial (HAE) cells [27,28]. The HAE cells from the lower respiratory tract possess mainly α-2,6- and α-2,3-linked NeuAc galactose receptors. Our studies showed that the H7N9 virus was able to infect all three cell lines from different origins, canine (MDCK), avian (CEF), and human (A549). This finding confirmed the findings that the H7N9 virus can bind to both types of sialic acid receptors [27,28]. Our findings suggested that the H7N9 virus behaved like an avian virus, despite the evidence that this H7N9 Anhui strain was isolated from an infected patient during the first wave of the epidemic in 2013. However, as the avian α-2,3-linked galactose NeuAc receptors were also found in the human bronchiolar and alveolar cells of the lower respiratory tract in human, the preference of H7N9 to the avian receptors also suggested that the H7N9 virus could bind to the lower respiratory tract to inflict severe manifestation in the same manner as that reported for the H5N1 virus [29,30].

We used microarray analysis to investigate the effect of H7N9 virus infection on each host cell transcriptome by comparing the host response with representative strains of LPAI virus and human virus subtypes. The H7N9 virus infection was able to induce a significant down-regulation of gene expression in A549 cells at 10 hpi, similar to what was observed with the human strain, H1N1-WSN (Figure 6A). In contrast, the H7N9 virus induced a distinct expression pattern in CEF cells at 10 hpi, compared to other human or avian virus subtypes. The findings in the current study showed that H7N9 virus could elicit a unique and stronger cytokine host gene expression response in A549 and CEF cells that is significantly different from the human H1N1 WSN strain and the LPAI H5 viruses (Figure 6A). This difference in findings included the virus subtype H9N2, which was reported to be responsible for the origins of six internal genes of the H7N9 virus [14,15,16]. The microarray data analysis was supported by qPCR performed on the CCL5 gene, which demonstrated almost an increase of 23 million-FC when normalized to the house-keeping gene, EF (Appendix A). The H7N9 virus host response profile resembled that of HPAI viruses, such as the H5N1 subtype, but not LPAI H9N2 subtype, and suggested that the cytokine storm could explain the reason behind the severity of human infection caused by the H7N9 virus infection.

In order to compare the gene expression profile between different host species, the avian genes were mapped to the human orthologs, as described in [21]. The functional annotation of the ortholog genes confirmed that Type 1 Interferon pathway (TGF-β signaling pathway) is the common proviral pathway activated in both H7N9 virus-infected human (A549) and avian (CEF) cells (Figure 5C). In addition, Type III Interferon pathway was also shown to be activated in H7N9 virus-infected A549 cells, and this was confirmed by the qPCR which detected the presence of IL28a in 790-FC when normalized to the house-keeping gene, EF (Appendix A). The interferon induction also led to the phosphorylation of STAT protein to pSTAT, and this may, in turn, have triggered and expressed other Interferon-stimulated genes (ISGs). QPCR was able to detect significant FC in ISG involved in antiviral activities, OAS2, RSAD2, and MX1 (Appendix A). In this case, the H7N9 virus behaves like the LPAI virus reported in our previous study [20], where both the Type I and Type III Interferon pathways can be activated in the human A549 cells.

The current study also identified an additional proviral signaling pathway, the HIPPO signaling pathway, that was unique to H7N9 activation only in CEF cells (Figure 3D, Figure 4D and Figure 5C). We did not observe this pathway when we compared the DEGs in the microarray data with other avian influenza subtypes (Figure 6). This is the first study that reported the HIPPO signaling pathway in influenza A virus infections. The HIPPO signaling pathway contains a cascade of threonine and serine kinases that regulates a number of growth and developmental physiological processes in controlling organ size by regulating homeostasis, and regenerations in mammalian hosts [31]. This pathway was named after one of the key component, protein kinase Hippo (Hpo) whose gene mutations lead to tissue overgrowth, resulting in a hippopotamus phenotype. Dysfunction of the HIPPO signaling pathway has been implicated in human diseases, including cancer [32]. It was first discovered in Drosophila in 2003, but later on found to be greatly conserved in mammalian cells and responsible for cell proliferation and apoptosis [33]. The YES associated protein (YAP), a major downstream effector of the HIPPO signaling pathway, was reported to be associated with cancer-causing viral proteins, such as the Hepatitis B virus X protein [34,35], and murine polyomavirus small T antigen [36]. Increased level of YAP protein was also found accumulating in the nucleus of human papillomavirus-positive oropharyngeal squamous cell carcinoma [37], whereas, the reverse was reported for hepatocellular carcinomas caused by hepatitis C [38]. The HIPPO pathway was also reported to be involved in the modulation of antiviral response by other virus infections, e.g., Sendai virus [39]. Recently, it was reported to be implicated in Zika-virus induced microcephaly [40]. These reported results suggested that the involvement of the HIPPO signaling pathway during virus infection can either lead to an overactive induction resulting in cancer, or cross-talk with other signaling cascade to maintain the virus in an antiviral state.

Severe influenza manifestation in human was reported to be generally associated with the induction of expression genes for inflammatory cytokines and down-regulation of lipid metabolic pathways. When human bronchial epithelial cell line, Calu-3, was used to infect H7N9 Anhui strain and H5N1, more similarities were reported in the transcriptomic analysis than between H7N9 and H7N7 [41]. Down-regulation of antigen presentation was also reported, and this seemed to be a characteristic associated with avian viruses. In mice studies, where BALB/c mice were infected with H7N9 Anhui strain, increased induction of cytokine genes, and decreased transcription of lipid metabolism and coagulation signaling were reported [42]. In our study, both H7N9 virus-infected A549 and CEF cells down-regulated several common cellular metabolic pathways, but with different magnitude (Appendix A). For example, metabolic pathways involving carbon metabolism, N-Glycan biosynthesis, purine metabolism, fatty acid metabolism, protein processing in the endoplasmic reticulum, cell cycle and glycolysis, were down-regulated suggesting that the host processes involving carbon and amino acid synthesis were actively used for virus replication in both cell lines. Other common metabolic pathways involving extracellular matrix (ECM) receptor, adherens junction, and focal adhesion, were also down-regulated, again an indication of active virus replication where the H7N9 virus continued to draw resources from the host in virus-infected cell lines. These observations confirmed the data reported by other studies where ECM pathways were significantly down-regulated in HPAI H7N9 and HPAI H5N1 [43]. A separate report also described that in human survivors of severe H7N9 infection, related proteins from the hydrolysis of fibronectin and collagens IV were found present in their plasma, suggesting that the ECM pathway was actively involved in remodeling damaged tissues [44].

## 5. Conclusions

Influenza A virus uses three mechanisms to inflict severe reactions in the human host (See review by [45]). The virus can either induces direct viral pathology, utilizing its innate immune response by up-regulating cytokine and damaging the ECM, followed by the cellular immune response when the cytokine storm is induced. Our studies have shown that H7N9 infections can induce the up-regulation of cytokines and cytokine-related gene expression in both human and avian cells. This cytokine storm is the same mechanism underlying the severe manifestation of H5N1 in humans. However, during avian infection by the H7N9 virus, the HIPPO signaling pathway induces the up-regulation of cancer pathways and down-regulation of metabolic processes, e.g., ECM at the proviral state, maintaining the virus in a self-defense state to repair any tissue damages induced by the cytokines; hence dampening the cytokine storm. We hypothesize that the cross-talk of the HIPPO pathways with other signaling pathways allows the H7N9 virus to remain in its low pathogenicity form in the avian host and this may explain the observation of a non-diseased state during the epidemic of H7N9.

## Figures and Tables

**Figure 1 cells-09-00448-f001:**
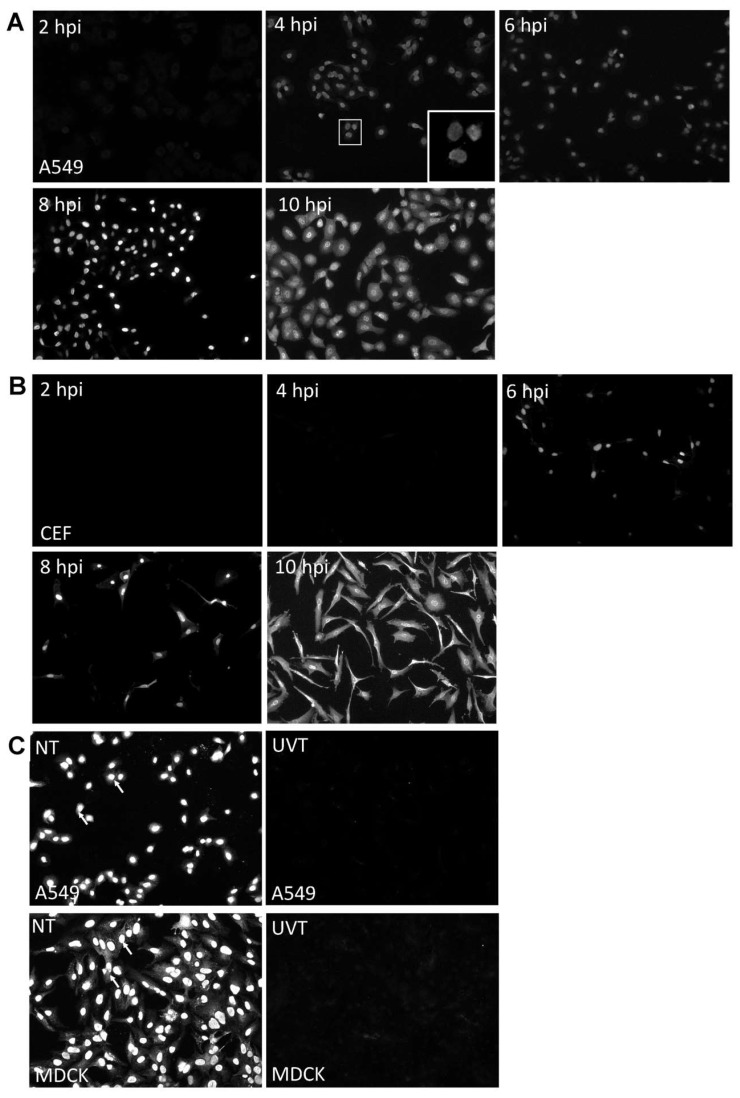
Examination of A549 cells, chick embryo fibroblast (CEF) and Madin-Darby canine kidney (MDCK) cells infected with H7N9 virus. (**A**) A549 cells and (**B**) CEF cells were H7N9 virus-infected at MOI of 5 and at 2,4,6,8 and 10 hours post-infection (hpi). The infected cells were stained with anti-influenza A virus and examined using immunofluorescence (IF) microscopy (objective × 40 magnification). The insert showing the virus-infected A549 cells at 4 hpi is an image at higher magnification. (**C**) A549 and MDCK cells were infected with untreated (NT) and UV-treated H7N9 virus (UVT) at MOI of 5. At 10 hpi, all cells were stained with anti-influenza A virus and examined using IF microscopy (objective × 40 magnification).

**Figure 2 cells-09-00448-f002:**
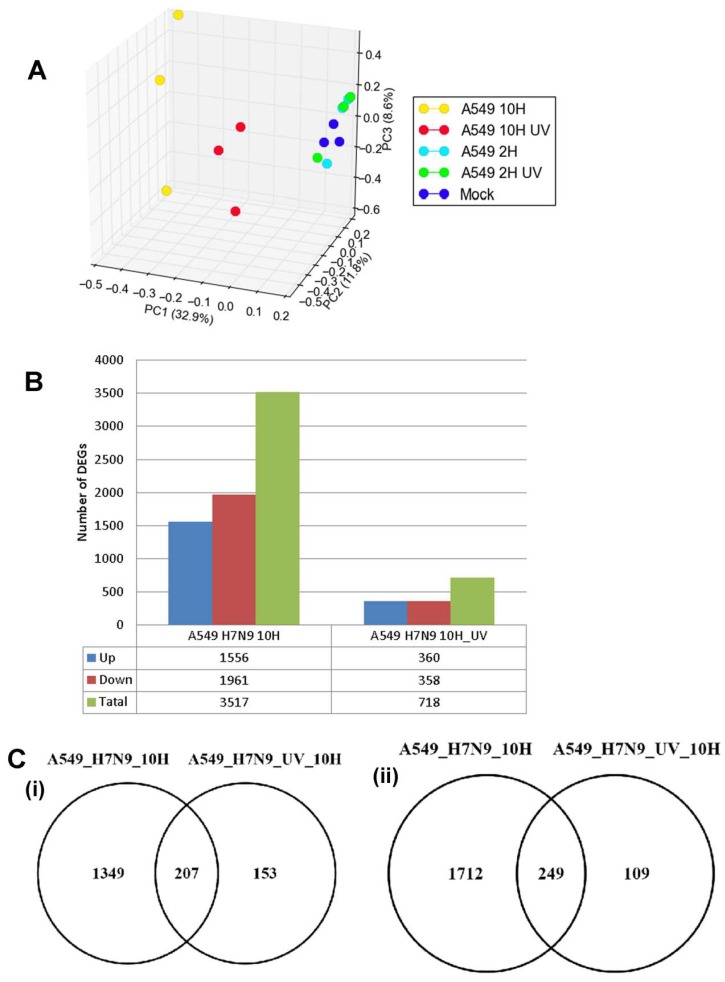
Differentially expressed genes (DEGs) in A549 cells infected with H7N9 virus. (**A**) Principal component analysis of the gene expression changes at 2 and 10 hours post-infection (hpi). (**B**) Number of up- and down-regulated genes in untreated and ultra-violet (UV)-treated virus infections. (**C**) Pairwise analysis of the DEGs at 10 hpi showing (i) up- and (ii) down-regulated genes in untreated and UV-treated virus infections. (**D**) Functional annotation of up-regulated DEGs. (**E**) Functional annotation of down-regulated DEGs.

**Figure 3 cells-09-00448-f003:**
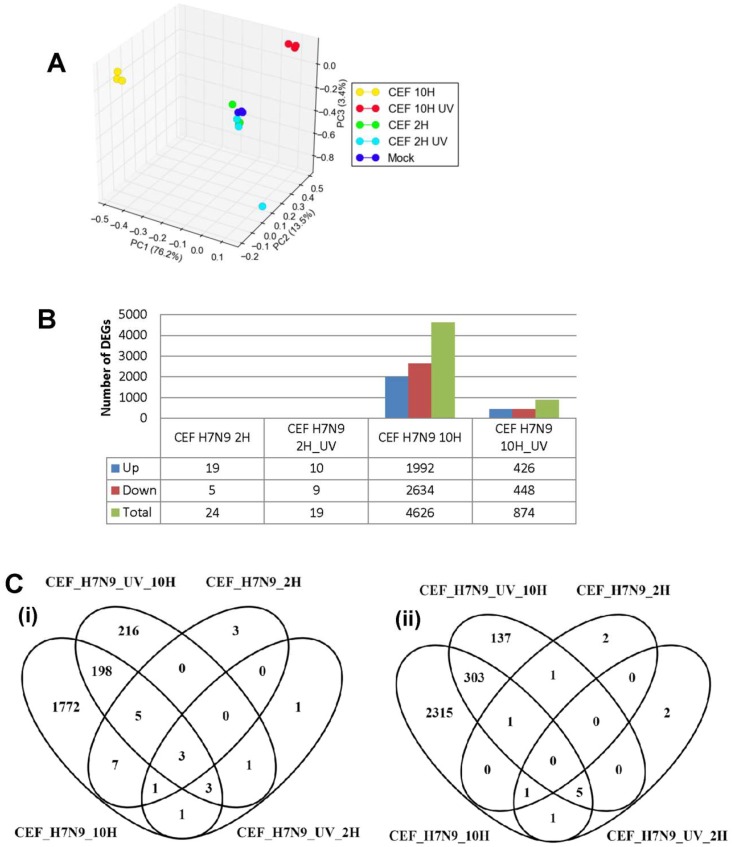
Differentially expressed genes (DEGs) in CEF cells infected with H7N9 virus. (**A**) Principal component analysis of the gene expression changes at 2 and 10 hours post-infection (hpi) (**B**) Number of up- and down-regulated genes in untreated and ultra-violet (UV)-treated virus infections. (**C**) Pairwise analysis of the DEGs at 10 hpi showing (i) up- and (ii) down-regulated genes in untreated and UV-treated virus infections. (**D**) Functional annotation of up-regulated DEGs. (**E**) Functional annotation of down-regulated DEGs.

**Figure 4 cells-09-00448-f004:**
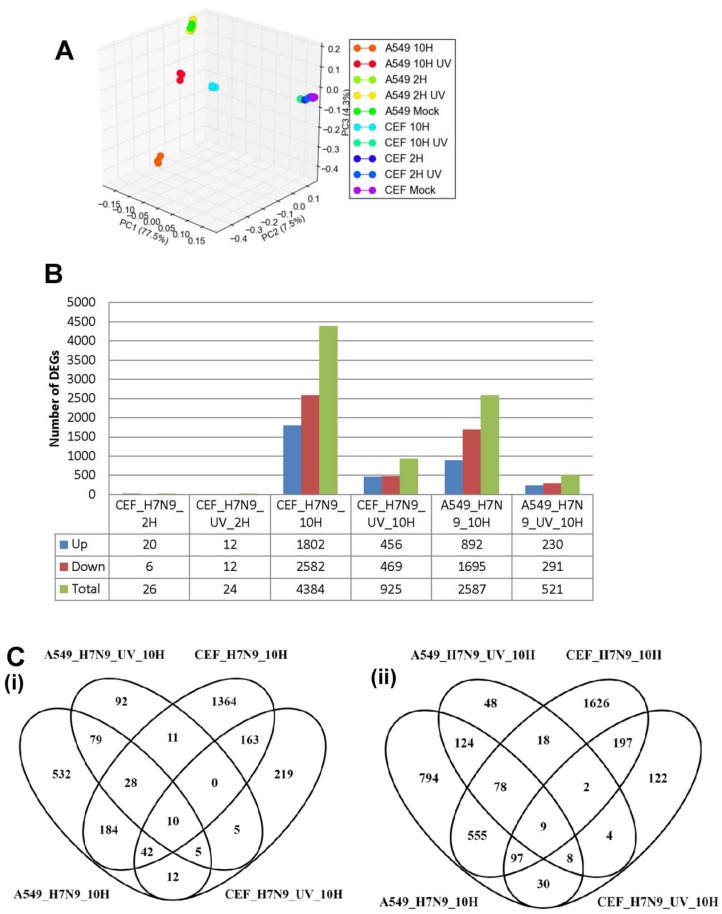
Comparison of orthologous differentially expressed genes (DEGs) in A549 and CEF cells infected with H7N9 virus. (**A**) Principal component analysis of the gene expression changes at 2 and 10 hours post-infection (hpi). (**B**) Number of up- and down-regulated orthologus genes in UV-treated and untreated H7N9 virus infections. (**C**) Pairwise analysis of the DEGs at 10 hpi showing (i) up- and (ii) down-regulated genes untreated and UV-treated virus infections. (**D**) Functional annotation of up-regulated orthologous DEG. (**E**) Functional annotation of down-regulated orthologous DEGs.

**Figure 5 cells-09-00448-f005:**
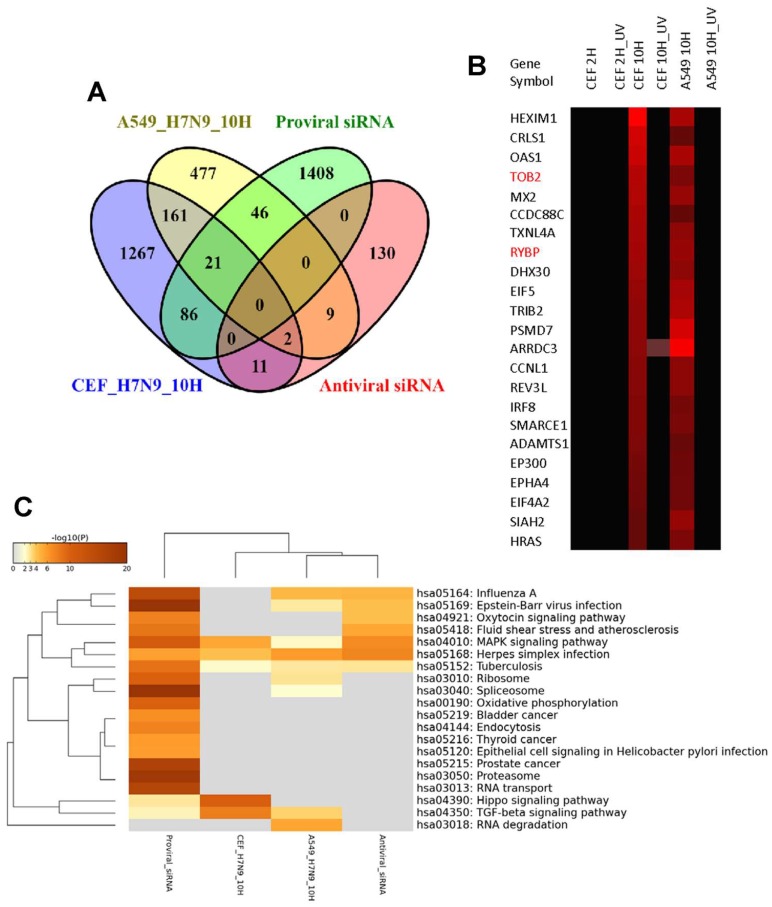
Integration of proviral and antiviral host factors (genes) with the orthologous differentially expressed genes (DEGs) in A549 and CEF cells infected with H7N9 virus. (**A**) Pairwise analysis of the DEGs with proviral and antiviral host factors from siRNA screening studies. (**B**) Heat map representing 21 proviral and 2 antiviral host factors (gene symbols highlighted in red) up-regulated in both virus-infected A549 and CEF cells. (**C**) Functional annotation of proviral and antiviral host factors from siRNA studies in line with up-regulated DEGs in virus-infected A549 and CEF cells at 10 hpi.

**Figure 6 cells-09-00448-f006:**
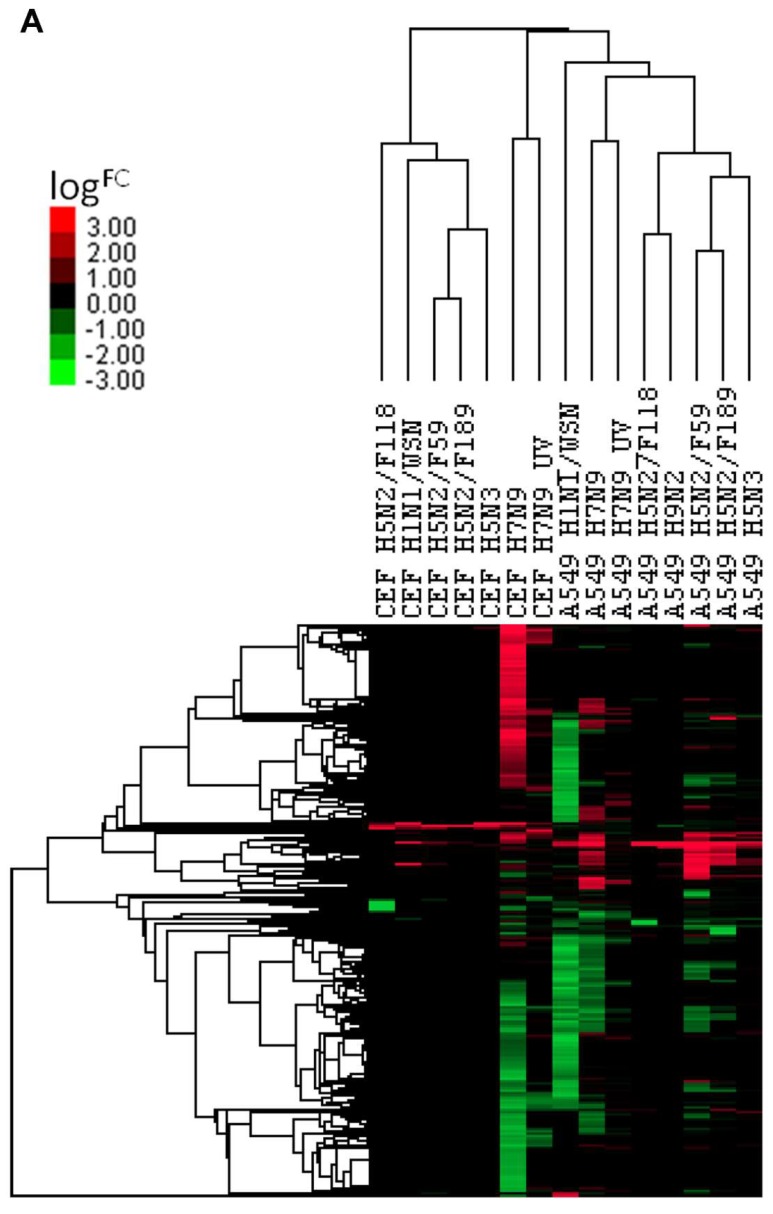
Comparison of orthologous differentially expressed genes for CEF and A549 cells with different influenza A virus subtypes. (**A**) A heat map representing hierarchical clustering of up-regulated and down-regulated DEGs. (**B**) and (**C**) Comparison of the KEGG pathways of the (**B**) up-regulated and (**C**) down-regulated genes.

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
