# Peer review of "A System Based-Approach to Examine Host Response during Infection with Influenza A Virus Subtype H7N9 in Human and Avian Cells"

_cells, 2020, doi:10.3390/cells9020448_

Round 1

Reviewer 1 Report

Taye et al present a research paper entitled “A system based-approach to examine host response 2 during infection with influenza A virus subtype 3 H7N9 in human and avian cells”, where they perform a system-based study of the infection of human and avian cell by the influenza A virus H7N9 subtype

The manuscript is generally well written and clear and presents an adequate experimental setup, appropriate controls and analyses which support the authors conclusions.

The following issues should be addressed prior to publication:

1- The figures have very low resolution. It was sometimes very hard to understand their contents and, hence, judge their results.

2- Figure 1 – Scale bars are missing. The authors claim that at 4hpi NP is localized in the nuclei, although no nuclear staining is shown. Images should be more amplified in order to clearly observe the NP cellular localization throughout the time points in both cells. In C, the y axis should contain a more complete description.

3- The reasoning for the use of MDCK cells should be present at the beginning of the results section.

4- Figure 2 – Legend of A is cut. The different panels should be better adjusted.

5 – Figure 3 – The data for the non-treated virus and the UV-treated virus are presented in a different orientation when compared to figure 1 E. For clarity and easier comparison between the results this should be uniformized.

6- Figure 6 – The color code bar is missing on panel A.

7- The references of all the used antibodies should be mentioned

8- line 19 – “maintained” should be substituted by “maintains”

9- line 41 – “protein” should be substituted by “proteins”

10- line 89 – “emerged” should be substituted by “emergence”

11- line 97 – “details” should be substituted by “detail”

12- line 223 – “quantitated” should be substituted by “quantified”

13 - Figure S1 misses a more complete legend.

Reviewer 2 Report

The study analyses influenza A virus (H7N9) infection in human and avian cell lines, assessing host response based on molecular assays. There are minor corrections regarding writing, and some comments. All corrections and comments were done in the pdf file (attached) using the "comment" tools.
